# A Clot Waveform Analysis Showing a Hypercoagulable State in Patients with Malignant Neoplasms

**DOI:** 10.3390/jcm10225352

**Published:** 2021-11-17

**Authors:** Mayu Kobayashi, Hideo Wada, Shunsuke Fukui, Hiroki Mizutani, Yuhuko Ichikawa, Katsuya Shiraki, Isao Moritani, Hidekazu Inoue, Motomu Shimaoka, Hideto Shimpo

**Affiliations:** 1Mie Prefectural General Medical Center, Department of Gastroenterology, Yokkaichi 510-0885, Japan; xsvaria1010@gmail.com (M.K.); shunsuke-fukui@mie-gmc.jp (S.F.); n279828658b79ku@gmail.com (H.M.); isao-moritani@mie-gmc.jp (I.M.); hidekazu-inoue@mie-gmc.jp (H.I.); 2Mie Prefectural General Medical Center, Department of General and Laboratory Medicine, Yokkaichi 510-0885, Japan; katsuya-shiraki@mie-gmc.jp; 3Mie Prefectural General Medical Center, Department of Central Laboratory Medicine, Yokkaichi 510-0885, Japan; ichi911239@yahoo.co.jp; 4Department of Molecular Pathobiology and Cell Adhesion Biology, Mie University Graduate School of Medicine, Tsu 514-8507, Japan; motomushimaoka@gmail.com; 5Mie Prefectural General Medical Center, Yokkaichi 510-0885, Japan; hideto-shimpo@mie-gmc.jp

**Keywords:** CWA, APTT, sTF/FIXa, cancer, thrombosis

## Abstract

(1) Objective: hypercoagulability in patients with malignant neoplasm were evaluated to examine the relationship with thrombosis. (2) Methods: clot waveform analysis (CWA)—activated partial thromboplastin time (APTT) and CWA—small amount of tissue factor induced FIX activation (sTF/FIXa) assays were performed in 92 patients with malignant neoplasm and the relationship between hypercoagulability and thrombosis was retrospectively examined. (3) Results: The study population included 92 patients with malignant neoplasms. Twenty-six (28.3%) had thrombotic diseases and 9 (9.8%) patients died within 28 days after the CWA. The peak time of the CWA-APTT could not show hypercoagulability in patients with malignant neoplasms. There were almost no significant differences in the peak times of the sTF/FIXa among patients with malignant neoplasms and healthy volunteers. In contrast, the peak heights of the CWA-sTF/FIXa in patients with various malignant neoplasms were significantly higher than those in healthy volunteers. Furthermore, among patients with malignant neoplasms, the peak heights of the sTF/FIXa in patients with thrombosis were significantly higher than those in patients without thrombosis. (4) Conclusions: although the routine APTT cannot evaluate the hypercoagulability, the peak heights of CWA-sTF/FIXa were significantly high in patients with malignant neoplasms, especially in those with thrombosis, suggesting that an elevated peak height of the CWA-sTF/FIXa may be a risk factor for thrombosis.

## 1. Introduction

Cancer associated thrombosis [1,2] has been recently attracted attention by many physicians and researchers. The incidence of venous thromboembolism in patients with cancer is 0.5%, while that in the general population is 0.1% and active cancer accounts for 20% of the overall incidence of venous thromboembolism [3]. Thrombosis is a leading cause of non-cancer death in cancer patients and includes both arterial and venous thromboembolism [4]. Although venous thromboembolism is considered one of the most important thrombotic diseases in cancer patients [1,5], Trousseau syndrome [6] is well-known thrombotic syndrome in cancer patients. This syndrome is first noted that unexpected or migratory thrombophlebitis could be a forewarning of an occult visceral malignancy [7]. Thereafter, this syndrome is extended as cancer-associated thrombosis, including acute cerebral infarction, venous thromboembolism and chronic disseminated intravascular coagulation (DIC) due to hypercoagulability [6,8,9,10,11]. Cancer-associated thrombosis is a heterogenous disease and various factors, including leukocytosis, thrombocytosis, several physical factors, and increased coagulation factors have been proposed as risk factors for thrombosis [1]. Hypercoagulability, including increased tissue factor (TF), which activates the extrinsic pathway of the coagulation system and TF-positive microparticles, was proposed as a risk factor for cancer-associated thrombosis [1]. Activation of the coagulation cascade through tumor growth and metastasis may cause thrombosis in cancer [12]. Platelet activation is also proposed as the mechanism of cancer-associated thrombosis [13]. Therefore, an early diagnosis and prophylaxis for thrombosis may be important in patients with cancer [6]. 

Although elevated D-dimer is useful for detecting thrombosis in patients with cancer [14,15], there are no routine test for hypercoagulability. Recently, a clot waveform analysis (CWA) to activated partial thromboplastin time (APTT) has been developed to analyze to evaluate hemostatic abnormalities in patients with bleeding tendency and to monitor anticoagulant therapy [16,17]. Furthermore, the use of the small amount of TF induced FIX activation assay (sTF/FIXa) assay can evaluate the hemostatic abnormalities including platelet abnormalities [18].

In this study, hypercoagulability in patients with malignant neoplasms was investigated using the CWA-APTT and sTF/FIXa and the relationship between the results of these tests and thrombosis was examined.

## 2. Materials and Methods

The study population included and 92 patients with the following conditions who were managed at Mie Prefectural General Medical Center from 21 August 2020 to 20 August 2021; hepatocellular carcinoma (n = 23); colon cancer (n = 4); stomach cancer (n = 8); prostate cancer (n = 5); biliary tract cancer (n = 7); lung cancer (n = 6); pancreatic cancer (n = 10); esophageal cancer (n = 4); malignant lymphoma (n = 14); myelodysplastic syndrome (n = 10); other cancer (n = 5; origin unknown, n = 2 melanoma, n = 1; uterus cancer, n = 1 and ovarian cancer, n = 1), including hepatocellular carcinoma and colon cancer (n = 1); stomach and prostate cancer (n = 1); stomach and biliary tract cancer (n = 1), stomach and lung cancer (n = 1). The study involved 18 healthy volunteers (8 males and 12 females; 21 to 56 years old).

DIC was diagnosed using the Japanese Ministry of Health Labor and Welfare criteria for DIC [19]. Acute cerebral infarction was diagnosed using computed tomography or magnetic resonance imaging, acute coronary syndrome was diagnosed using coronary angiography, electrocardiography and was based on elevated troponin levels, venous thromboembolism was diagnosed using computed tomography or venous Doppler ultrasound. In the CWA-APTT and sTF/FIXa, 20 healthy volunteers (12 females and 8 males; median age, 35.5 years old; 25–75 percentile, 33.0–35.0 years old) were also examined as control.

The study protocol (2019-K9) was approved by the Human Ethics Review Committee of Mie Prefectural General Medical Center, and informed consent was obtained from each participant. This study was carried out in accordance with the principles of the Declaration of Helsinki.

The CWA-APTT was performed using APTT-SP^®^, which uses silica as an activator of FXII and synthetic PLs (Instrumentation Laboratory; Bedford, MA, USA) and platelet poor plasma, with an ACL-TOP^®^ system (Instrumentation Laboratory) as previously reported [16,20]. The CWA-sTF/FIXa assay was performed using platelet rich plasma and 2000-fold diluted HemosIL RecombiPlasTin 2G (TF concentration < 0.1 pg/mL; Instrumentation Laboratory). Three curves are shown on the monitor of this system [11]. One curve shows the changes in the absorbance observed while measuring the APTT, corresponding to fibrin formation (FF, navy line). The second (first derivative peak, 1st DP; red line), corresponds to the coagulation velocity, and the third (second derivative peak, 2nd DP; light blue) corresponds to the coagulation acceleration. The height and time of the 1st DP, 2nd DP and FF are called the 1st DP height (1st DPH) and 1st DP time (1st DPT), 2nd DPH and 2nd DPT, and FFH and FFT, respectively (Figure 1). Platelet rich plasma was prepared by centrifugation at 900 rpm for 15 min (platelet count, 40 × 10^10^/L), and platelet-poor plasma was prepared by centrifugation at 3000 rpm for 15 min (platelet count, <0.5 × 10^10^/L) [18].

### Statistical Analyses

The data are expressed as the median (range). The significance of differences between groups was examined using the Mann–Whitney U-test. *p* values of <0.05 were considered to indicate a statistical significance. All statistical analyses were performed using the Stat-Flex software program (version 6; Artec Co Ltd., Osaka, Japan). 

## 3. Results

Figure 1 shows that there were no significant differences in the peak time of the APTT among a healthy volunteer, pancreatic cancer patient without and with acute cerebral infarction and non-cancer patient with acute cerebral infarction. Although the peak heights of the CWA-APTT were significantly higher in these three patients than healthy volunteer, there was no significant difference in the peak heights of the APTT among the above three patients. The CWA-sTF/FIXa assay shows that the peak heights were markedly higher in pancreatic cancer patient with acute cerebral infarction than in that without, whose levels were higher than those of healthy volunteer. Table 1 shows 92 patients with malignant neoplasms, 26 (28.3%) of whom had associated thrombotic diseases (12 peripheral artery thrombosis and venous thromboembolism, n = 12; acute cerebral infarction, n = 7; acute coronary syndrome, n = 4; DIC, n = 2; and thrombotic microangiopathy, n = 1). Peripheral artery thrombosis and venous thromboembolism included deep vein thrombosis (n = 5), portal vein thrombosis (n = 4) and pulmonary.

Embolisms (n = 2) and peripheral artery thrombosis (n = 1). Nine patients (9.4%) died within 28 days after the CWA examination. 

Table 2 shows the peak time and height of the CWA-APTT in patients with various malignant neoplasms. The three peak times of the CWA-APTT were significantly longer in patients with stomach cancer, biliary tract cancer, pancreatic cancer, esophageal cancer, times of the sTF/FIXa among patients with various malignant neoplasms and healthy volunteers. In contrast, the peak heights of the CWA-sTF/FIXa in patients with hepatocellular carcinoma, colon cancer, stomach cancer, prostate cancer, lung cancer, pancreatic cancer, esophageal cancer, and malignant lymphoma were significantly higher in comparison to those in healthy volunteers (Table 3).

Regarding the relationship between the CWA and thrombosis, patients treated with anticoagulants were excluded. There were no significant differences in the three peak times of the CWA-APTT among healthy volunteers, and malignant neoplasm patients with or without thrombosis (Figure 2a). The three peak heights of the CWA-APTT were significantly higher in malignant neoplasm patients with or without thrombosis than in healthy volunteers. FFH was the only parameter that was significantly higher in those with thrombosis than in those without (Figure 2b). The three peak times of the CWA-sTF/FIXa were significantly longer in malignant neoplasm patients with or without thrombosis in comparison to healthy volunteers (Figure 3a). The three peak heights of the CWA-sTF/FIXa were significantly higher in those patients with or without thrombosis than in healthy volunteers. The 2nd DPH and 1st DPH of the CWA-sTF/FIXa were significantly higher in those patients with thrombosis than in those without thrombosis (Figure 3b).

## 4. Discussion

In this study, 28.3% of patients with malignant neoplasm had thrombotic diseases, suggesting that these patients were in a hypercoagulable state. Patients with malignant neoplasm are frequently associated with thrombotic disorder and are considered to be in a hypercoagulable state [21]. Trousseaus syndrome has been shown in multiple published articles [22,23,24,25,26] to have a relationship between malignant neoplasms and thrombotic disease [6]. The existence of multiple definitions of Trousseau’s syndrome, or cancer- associated thrombosis, is due in part to the existence of multiple pathophysiologic mechanisms that apparently contribute to the hypercoagulability associated with cancer [6]. The mechanism underlying cancer-associated thrombosis reportedly involves the over-expression of TF [22], cysteine proteinase [23], tumor hypoxia [24], carcinomas mucins [25], and oncogene activation [26]. These risk factors overlap to cause thrombosis in patients with cancer [6]. Although the specific relationship between the type of neoplasm and thrombotic diseases is not clear due to the small size of this study, cases of stomach cancer and hematological malignancy with DIC [27,28], hepatocellular carcinoma with portal vein thrombosis [29,30], and pancreatic cancer and brain tumor with venous thromboembolism [31] have been previously reported. As cancer-associated thrombosis causes a fatal hypercoagulable state, anticoagulation therapy should be administered to these patients. Therefore, the early detection of a hypercoagulable state may be important.

Routine assays as APTT and prothrombin time are generally used for monitoring anticoagulant or diagnosing bleeding disorders, but they are not used to evaluate the hypercoagulability. Elevated D-dimer levels are useful to suspect venous thromboembolism [32] but cannot evaluate the hypercoagulability. Although thrombin generation test or thromboelastography may detect the hypercoagulability, these tests are not routine assay. The peak height of 1st and 2nd DP of APTT was reported to able to evaluate hemostatic ability [16]. The present study showed that the peak times of APTT which correspond routine APTT, could not evaluate the hypercoagulability in patients with malignant neoplasm. Although the peak height of APTT was significantly high in patients with malignant neoplasm, there were no significant difference in the peak height of APTT between these patients with and without thrombosis. Therefore, APTT included the activator for contact factors and massive phospholipid, suggesting that APTT cannot show a physiological coagulation ability [16,18].

The CWA-sTF/FIXa assay uses plalet-rich plasma as physiological phospholipid which does not activate contact factor and reflects platelet activation [16,18]. The prolongation of peak times in patients with malignant neoplasm reflect with anticoagulants. The prolongation of peak time in CWA-APTT and sTF/FIXa due to anticoagulant were previously reported in major orthopedic surgery [17]. The peak time CWA-APTT or -sTF/FIXa may reflect with anticoagulant, and the peak height may reflect physiological clotting activity. The peak heights of CWA-sTF/FIXa show not only hypercoagulability in patients with malignant neoplasm but also possibility of prediction of thrombosis. One case (Figure 1(I-c,II-c)) showed markedly elevated peak height before onset of acute cerebral infarction. This case should be treated with anticoagulant.

## 5. Conclusions

Patients with malignant neoplasm are in a hypercoagulable state and often associated with thrombosis. The peak height of CWA-APTT and sTF/FIXa was extremely high in patients with malignant neoplasm, especially those with thrombosis, suggesting that the peak height of CWA-sTF/FIXa may be useful to detect the hypercoagulability.

## Figures and Tables

**Figure 1 jcm-10-05352-f001:**
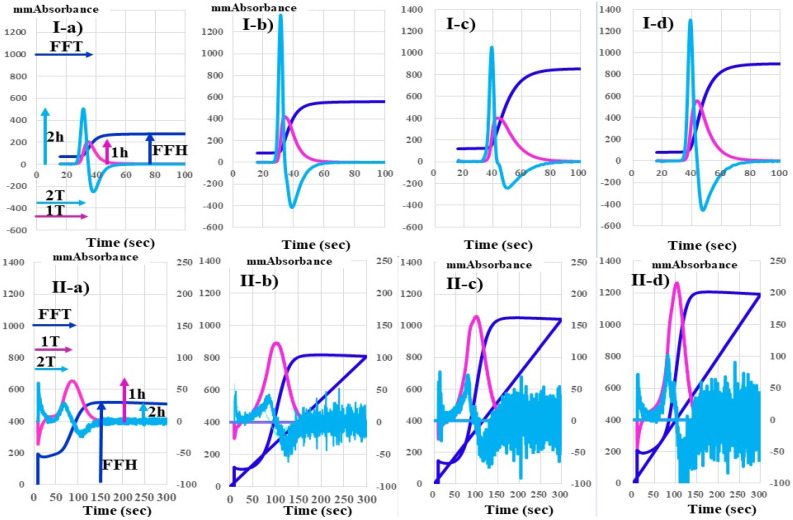
CWA-APTT (**I**) and CWA-sTF-FIXa (**II**) in a healthy volunteer (**a**), pancreatic cancer patient without thrombosis (**b**), pancreatic cancer patient before ACI (**c**) and patient with ACI (**d**). APTT, activated partial thromboplastin time; sTF/FIXa, small tissue factor induced FIX activation assay; ACI, acute cerebral thrombosis; navy line, fibrin formation curve; red line, 1st derivative curve (velocity); light-blue line, 2nd derivative curve (acceleration); 1T, 1st derivative peak time; 2T, 2nd derivative peak time; FFT, fibrin formation time; 1 h, 1st derivative peak height; 2 h, 2nd derivative peak height; FFH, fibrin formation height, TM (mm abs); transmittance (mm absorbance).

**Figure 2 jcm-10-05352-f002:**
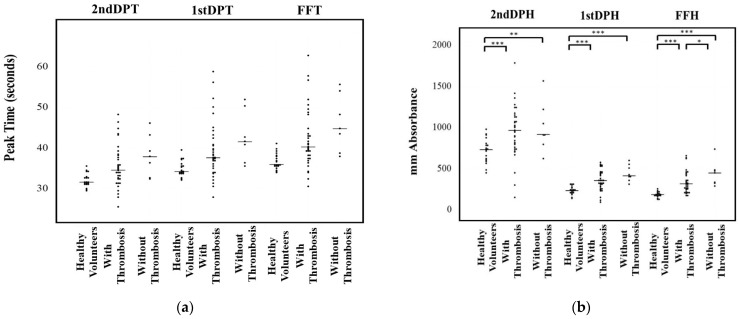
Peak time (**a**) and height (**b**) of activated partial thromboplastin time. DPT, derivative peak time; FFT, fibrin formation time; DPH, derivative peak height; FFH, fibrin formation height; *, *p* < 0.05; **, *p* < 0.01; ***, *p* < 0.001.

**Figure 3 jcm-10-05352-f003:**
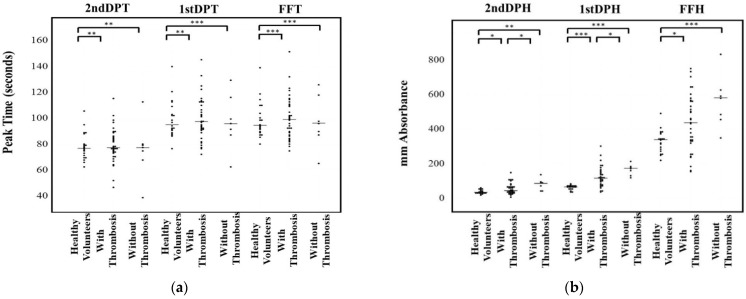
Peak time (**a**) and height (**b**) of small amount of tissue factor induced FIX activation assay. DPT, derivative peak time; FFT, fibrin formation time; DPH, derivative peak height; FFH, fibrin formation height; *, *p* < 0.05; **, *p* < 0.01; ***, *p* < 0.001.

**Table 1 jcm-10-05352-t001:** Subjects.

Disease	N	Age	Male	PAVTE	ACI	ACS	DIC	TMA	Mortality
Hepatocellular carcinoma *	23	73.4 ± 6.9	18	3	2	1	0	0	0 (0%)
Colon cancer *	4	74.8 ± 11.2	3	1	0	0	0	0	0 (0%)
Stomach cancer ^#,&,+^	8	76.6 ± 9.6	8	0	1	0	1	1	3 (37.5%)
Prostate cancer ^#^	5	80.4 ± 5.9	5	1	1	0	0	0	1 ^#^ (20.0%)
Biliary tract cancer ^&^	7	79.3 ± 6.4	3	1	0	0	0	0	1 (14.3%)
Lung cancer ^+^	6	70.3 ± 11.0	6	0	0	0	0	0	1 ^+^ (16.7%)
Pancreatic cancer	10	67.6 ± 10.3	7	1	1	0	0	0	2 (20.0%)
Esophageal cancer	4	65.8 ± 10.7	4	0	1	0	0	0	0 (0%)
Malignant lymphoma	14	77.4 ± 6.0	8	2	1	2	0	0	1 (7.1%)
Myelodysplastic syndrome	10	78.2 ± 11.3	4	1	0	0	1	0	0 (0%)
Others	5	67.7 ± 9.4	2	2	0	1	0	0	0 (0%)
Total	96	73.9 ± 9.3	68	12	7	4	2	1	9 (9.4%)

*, a patient with hepatocellular carcinoma and colon cancer; ^#^, a patient with stomach and prostate cancer; ^&^, a patient with stomach and biliary tract cancer; ^+^, a patient with stomach and lung cancer; PAVTE, peripheral arterial and venous thromboembolism, ACI, acute cerebral infarction; ACS, acute coronary syndrome; DIC, disseminated intravascular coagulation; TMA, thrombotic microangiopathic anemia.

**Table 2 jcm-10-05352-t002:** CWA-APTT in patients with malignant neoplasm.

	2nd DPT	1st DPT	FFT	2nd DPH	1st DPH	FFH
Cancer Type	Seconds			mm Absorbance		
Hepatocellular carcinoma	32.7 (26.8–42.0)	35.2 (28.9–46.1)	38.2 (31.1–53.4)	883 ** (264–1981)	283 * (139–801)	250 *** (164–759)
Colon cancer	33.5 (31.2–39.3)	36.7 (33.8–42.7)	40.4 (36.2–45.9)	1235 ** (1024–1452)	460 ** (330–509)	387 ** (286–447)
Stomach cancer	34.7 (28.7–38.5)	37.5 (31.5–42.4)	40.0 (34.0–49.5) *	773 (450–1278)	307 * (147–457)	279 *** (172–480)
Prostate cancer	34.8 (29.5–60.7)	37.7 (32.2–65.4)	39.9 (34.4–67.8)	810 (647–1220)	317 ** (254–427)	296 ** (214–330)
Biliary tract cancer	34.0 * (25.7–48.3)	37.0 * (28.0–58.8)	40.6 * (30.5–62.7)	784 (151–1789)	315 (90.7–544)	257 *** (221–621)
Lung cancer	34.0 (29.9–40.3)	36.6 (32.3–45.1)	39.5 (34.2–49.5)	752 (443–1004)	259 (132–450)	224 * (178–480)
Pancreatic cancer	35.2 *** (32.6–39.3)	38.5 *** (36.2–42.8)	41.4 *** (38.1–48.2)	973 * (302–1422)	353 ** (254–559)	336 *** (172–489)
Esophageal cancer	37.9 (31.4–44.7)	42.6 (34.1–50.0)	44.8 * (36.9–51.9)	1215 ** (972–1572)	491 ** (418–595)	448 ** (317–477)
malignant lymphoma	36.1 * (28.0–56.2)	39.7 * (30.5–60.5)	43.9 ** (32.5–68.7)	863 (274–1361)	311 ** (157–577)	323 *** (195–634)
Myelodysplastic syndrome	37.5 *** (31.5–78.5)	40.0 ** (34.1–88.5)	46.9 ** (35.8–90.3)	712 (128–1144)	337 (157–577)	256 *** (182–653)
Others	38.1 (31.1–44.5)	43.3 * (33.8–52.1)	44.9 * (35.8–55.6)	845 (596–1246)	328 (187–552)	319 (163–738)
Healthy volunteers	31.7 (29.6–35.7)	34.3 (32.1–39.7)	36.0 (34.0–41.1)	734 (453–975)	235 (137–316)	187 (131–256)

Data are shown as the median (range). CWA-APTT, clot waveform analysis-activated partial thromboplastin time; DPT, derivative peak time; DPH, derivative peak height; FFT, fibrin formation time; FFH, fibrin formation height; *, *p* < 0.05; **, *p* < 0.01; ***, *p* < 0.001 compared to healthy volunteers.

**Table 3 jcm-10-05352-t003:** CWA-sTF/FIXa in patients with malignant neoplasm.

	2nd DPT	1st DPT	FFT	2nd DPH	1st DPH	FFH
Cancer Type	Seconds			mm Absorbance		
Hepatocellular carcinoma	77.9 (52.2–105)	95.6 (76.4–130)	95.9 (78.2–127)	48.0 (21.0–108)	97.9 *** (49.9–270)	352 (205–938)
Colon cancer	83.2 (76.6–97.6)	101.5 (92.8–133)	102 (87.2–134)	68.9 (28.6–111)	139 ** (110–193)	543 * (334–639)
Stomach cancer	82.2 (63.9–148)	97.0 (81.9–207)	97.2 (82.4–196)	46.3 (11.4–102)	106 * (28.3–178)	370 (156–564)
Prostate cancer	72.7 (38.6–93.9)	91.8 (62.3–105)	89.6 (65.6–93.9)	46.4 (36.9–95.0)	80.9 * (64.8–178)	455 (257–934)
Biliary tract cancer	83.0 (46.3–148)	103 (79.3–207)	102 (83.1–196)	70.5 (13.8–109)	111 (28.3–250)	364 (184–702)
Lung cancer	72.6 (66.6–115)	82.6 (76.4–146)	84.9 (78.0–152)	49.1 (26.6–84.1)	92.0 (46.1–181) *	314 (183–645)
Pancreatic cancer	80.9 (70.6–102)	102 (91.2–133)	103 (88.7–132)	36.0 (19.0–72.6)	121 (46.2–165) **	411 (163–602) *
Esophageal cancer	74.8 (73.6–85.0)	98.7 (90.9–113)	98.9 (89.9–115)	69.8 (43.5–112) *	173 (123–213) **	600 (464–645) **
malignant lymphoma	76.1 (50.0–147)	93.1 (72.0–178)	92.0 (74.9–174)	45.6 (17.3–173)	101 (62.5–304) ***	437 (246–751) **
Myelodysplastic syndrome	90.5 (52.2–189)	113 (60.2–237)	111 (61.2–225)	58.6 (9.8–69.8)	91.4 (33.6–222)	329 (265–734)
Others	106 (73.6–132) *	129 (84.8–171)	126 (84.2–169)	38.9 (15.1–47.7)	69.6 (36.4–189)	406 (285–832)
Healthy volunteers	76.9 (62.5–106)	95.3 (76.7–140)	94.6 (80.3–139)	37.4 (21.2–60.9)	67.8 (37.2–88.3)	341 (221–495)

Data are shown as the median (range). CWA-sTF/FIXa, clot waveform analysis-small amount tissue factor induced FIX activation assay; DPT, derivative peak time; DPH, derivative peak height; FFT, fibrin formation time; FFH, fibrin formation height; *, *p* < 0.05; **, *p* < 0.01; ***, *p* < 0.001 compared to healthy volunteers.

## Data Availability

The data presented in this study are available on request to the corresponding author. The data are not publicly available due to privacy restrictions.

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
