# Peer review of "A Clot Waveform Analysis Showing a Hypercoagulable State in Patients with Malignant Neoplasms"

_jcm, 2021, doi:10.3390/jcm10225352_

Round 1

Reviewer 1 Report

An interesting study looking at the value of CWA-APTT and CWA-sTF/FIX in patients with malignancy. My comments are as follows:

1) The authors did not state when the test samples were collected in relation to the study. Were they prospectively collected after study recruitment? Did the recruitment take into account the thrombosis status of the patient or was it random. Such information will be useful to the reader.

2) In line 64, ‘managed at Mie Prefectural General Medical Center for the following conditions’ should read as ‘managed at Mie Prefectural General Medical Center:’

3) In line 174, ‘Patients with malignant neoplasm are frequently associated thrombotic disorders’ should read ‘are frequently associated with thrombotic disorders’

4) Will suggest a re-reading of the discussion and re-formatting, correcting for grammatical errors. Although, I can understand what the authors are trying to conclude, it is difficult for the casual reader to follow, especially with the grammatical and syntax errors in some of the sentences.

Author Response

Comment 1.                                                                      The authors did not state when the test samples were collected in relation to the study. Were they prospectively collected after study recruitment? Did the recruitment take into account the thrombosis status of the patient or was it random. Such information will be useful to the reader.

Response 1. This study was a retrospective study, which we have now stated in the Abstract and Materials and Methods.   

Comment 2.                                                                         In line 64, ‘managed at Mie Prefectural General Medical Center for the following conditions’ should read as ‘managed at Mie Prefectural General Medical Center:’

Response 2. In line 64, for the following conditions” was deleted.   

Comment 3.                                                            In line 174, ‘Patients with malignant neoplasm are frequently associated thrombotic disorders’ should read ‘are frequently associated with thrombotic disorders’

Response 3. This sentence was changed in accordance with the comment from reviewer.

Comment 4.                                                         Will suggest a re-reading of the discussion and re-formatting, correcting for grammatical errors. Although, I can understand what the authors are trying to conclude, it is difficult for the casual reader to follow, especially with the grammatical and syntax errors in some of the sentences.

Response 4. As suggested, we have now had a professional medical editor whose native language is English proofread the revised manuscript.

Reviewer 2 Report

Clot waveform analysis show hypercoagulable state in patients with 2 malignant neoplasm

The article proposes a new method of evaluating hypercoagulability in a routine, clinical setting. The authors emphasize the importance of assessing clot risk in cancer patients due to the presence of trousseaus syndrome. The core subject is relevant and interesting, but requires some polishing.

The level of English is lacking, we suggest the authors have their manuscript edited by either a native English speaker or third-party service. This will greatly improve the clarity, as in the results and discussion sections the authors tend to repeat themselves and make winding sentences that confuse the reader. Authors should also consider using less abbreviations to help the reader, as is, the current article is tedious because every sentence contains multiple abbreviations that must be looked up.

More specifically, the objective section in the abstract is not an objective but a method, please rephrase this. The introduction is too generalized, the authors do not explain what is Trousseaus syndrome and what this implies for cancer patients (hypercoagulable state) or how platelets and the coagulation cascade are implied/altered (tumor cell induced platelet aggregation, platelet degranulation, microvesicle generation, cancer or microvesicle tissue factor expression, etc.). Also, VTE is a consequence of trousseaus syndrome, not another unrelated syndrome.

We would like to ask if the venous ultrasound used to diagnose VTE was doppler, if so, please specify. We also believe that in this section, in the paragraph where they describe the APPT-CWA and CWA-sTF/FIX assays the authors should explain in more detail how the results are shown and depict each curve with its color code for easier interpretation of the figures. Incidentally, figure 1 should be larger and clearer, with a legend for the color codes of the curves; we also suggest that the authors re-organize this figure for clarity, as the labels for each graph are small and hard to read; making the interpretation of the data a difficult task.  Also, the bottom graphs have no legend for the light Y axis, please add this.

We also encourage the authors to rewrite their results section, especially for figure 1 as in the current form there are too many abbreviation and repetition. We suggest grouping the cancers into system types, or however the authors want but please make it clearer and easier to read and follow. Table 1 shows the patient characteristics but there is no mention of the healthy controls the authors studied, these should be included. Additionally, it is pancreatic cancer, no pancreas cancer. Again, in tables 2 and 3 please just write the cancer type instead of using abbreviations.

Figures 2 and 3 have an enormous amount of information; could the authors show the information in a more representative (and easier to analyze) format? We think that they are a little crowded and again, with too many abbreviations.

Finally, the discussion could be lengthened and supplemented with more references. For example, authors state in paragraph 1 :

“Although specific 178 relationship between malignant neoplasm and thrombotic diseases is not clear due to 179 small size study, stomach cancer with DIC [17] or HCC with portal vein thrombosis [18] 180 were previously reported”

However, it has been demonstrated in multiple published articles that there is a relationship between malignant neoplasms and thrombotic disease (AKA trousseaus syndrome) and that this is more prevalent in some cancer types like pancreatic, lung and brain tumors.

The second paragraph is the single most important paragraph as it explains the context and gives a solution to the proposed problem, please edit this paragraph to make the message clearer for the reader. Finally, the conclusion is missing a phrase stating the possible impact pf this research and future perspectives.

Author Response

Comment 1.                                                                 The level of English is lacking, we suggest the authors have their manuscript edited by either a native English speaker or third-party service. This will greatly improve the clarity, as in the results and discussion sections the authors tend to repeat themselves and make winding sentences that confuse the reader. Authors should also consider using less abbreviations to help the reader, as is, the current article is tedious because every sentence contains multiple abbreviations that must be looked up.

Response 1. As suggested, we have now had a professional medical editor whose native language is English proofread the revised manuscript. In addition, the number of abbreviations was decreased.

Comment 2.                                                               More specifically, the objective section in the abstract is not an objective but a method, please rephrase this. The introduction is too generalized, the authors do not explain what is Trousseaus syndrome and what this implies for cancer patients (hypercoagulable state) or how platelets and the coagulation cascade are implied/altered (tumor cell induced platelet aggregation, platelet degranulation, microvesicle generation, cancer or microvesicle tissue factor expression, etc.). Also, VTE is a consequence of trousseaus syndrome, not another unrelated syndrome.

Response 2. The objective section has now been revised, along with the Introduction. 

Comment 3.                                                                We would like to ask if the venous ultrasound used to diagnose VTE was doppler, if so, please specify. We also believe that in this section, in the paragraph where they describe the APPT-CWA and CWA-sTF/FIX assays the authors should explain in more detail how the results are shown and depict each curve with its color code for easier interpretation of the figures. Incidentally, figure 1 should be larger and clearer, with a legend for the color codes of the curves; we also suggest that the authors re-organize this figure for clarity, as the labels for each graph are small and hard to read; making the interpretation of the data a difficult task. Also, the bottom graphs have no legend for the light Y axis, please add this.

Response 3. In the Materials and Methods, "Doppler” has been added. In addition, the Materials and Methods and Figure legends have been revised, and Figure 1 was revised and enlarged.

Comment 4. We also encourage the authors to rewrite their results section, especially for figure 1 as in the current form there are too many abbreviations and repetition. We suggest grouping the cancers into system types, or however the authors want but please make it clearer and easier to read and follow. Table 1 shows the patient characteristics but there is no mention of the healthy controls the authors studied, these should be included. Additionally, it is pancreatic cancer, no pancreas cancer. Again, in tables 2 and 3 please just write the cancer type instead of using abbreviations.

Response 4. The results section has now been revised. Data from healthy volunteers were added to the Materials and Methods, and Tables 2 and 3 have been revised.

Comment 5.                                                             Figures 2 and 3 have an enormous amount of information; could the authors show the information in a more representative (and easier to analyze) format? We think that they are a little crowded and again, with too many abbreviations.

Response 5. Figures 2 and 3 have now been revised.

Comment 6.                                                             Finally, the discussion could be lengthened and supplemented with more references. For example, authors state in paragraph 1:

“Although specific 178 relationship between malignant neoplasm and thrombotic diseases is not clear due to 179 small size study, stomach cancer with DIC [17] or HCC with portal vein thrombosis [18] 180 were previously reported”

Response 6. The discussion has been fully revised, with added many new references added for support.

Comment 7.

The second paragraph is the single most important paragraph as it explains the context and gives a solution to the proposed problem, please edit this paragraph to make the message clearer for the reader.

However, it has been demonstrated in multiple published articles that there is a relationship between malignant neoplasms and thrombotic disease (AKA trousseaus syndrome) and that this is more prevalent in some cancer types like pancreatic, lung and brain tumors.

Response 7. The discussion has been fully revised, with many new references added for support.

Comment 8.                                                                Finally, the conclusion is missing a phrase stating the possible impact pf this research and future perspectives.

Response 8. The conclusion has now been revised.